# Regulatory Effect of *Bacillus subtilis* on Cytokines of Dendritic Cells in Grass Carp (*Ctenopharyngodon Idella*)

**DOI:** 10.3390/ijms20020389

**Published:** 2019-01-17

**Authors:** Chengchong Zhou, Hui Wang, Xige Li, Yaner Luo, Mengqi Xie, Zhixin Wu, Xiaoxuan Chen

**Affiliations:** 1Department of Aquatic Animal Medicine, College of Fisheries, Huazhong Agricultural University, Wuhan 430070, China; chengchongzhou@163.com (C.Z.); WangH9307@163.com (H.W.); xigeli@163.com (X.L.); luoyaner216@163.com (Y.L.); mq_xie@foxmail.com (M.X.); wuzhixin@mail.hzau.edu.cn (Z.W.); 2Hubei Engineering Technology Research Center for Aquatic Animal Diseases Control and Prevention, Wuhan 430070, China; 3Hubei Provincial Engineering Laboratory for Pond Aquaculture, Wuhan 430070, China

**Keywords:** dendritic cell, *Ctenopharyngodon idella*, Bacillus subtilis, cytokine, transcriptome

## Abstract

*Bacillus subtilis* is a common group of probiotics that have been widely used in the feed industry as they can increase host resistance to pathogens and balance the immune response. However, the regulatory mechanism of *Bacillus subtilis* on the host immune system remains unclear in teleosts. In this study, we isolated and enriched dendritic cells from white blood cells (WBCs), and then stimulated them with *Bacillus subtilis*. Morphological features, specific biological functions, and authorized functional molecular markers were used in the identification of dendritic cells. Subsequently, we collected stimulated cells at 0, 4, and 18 h, and then constructed and sequenced the transcriptomic libraries. A transcriptome analysis showed that 2557 genes were up-regulated and 1708 were down-regulated at 4 h compared with the control group (|Fold Change| ≥ 4), and 1131 genes were up-regulated and 1769 were down-regulated between the cells collected at 18 h and 4 h (|Fold Change| ≥ 4). Gene Ontology (GO) annotations suggested many differentially expressed genes (DEGs) (*p* < 0.05 and |Fold Change| ≥ 4) were involved in immune-related biological functions including immune system progress, cytokine receptor binding, and cytokine binding. The Kyoto Encyclopedia of Genes and Genomes (KEGG) pathway analysis showed that the cytokine–cytokine receptor interaction pathways were significantly enriched at both time points (*p* < 0.05), which may play a key role in the response to stimulation. Furthermore, mRNA expression level examination of several pro-inflammatory cytokines and anti-inflammatory cytokines genes by quantitative real-time polymerase chain reaction (qRT-PCR) indicated that their expressions can be significantly increased in *Bacillus subtili*, which suggest that *Bacillus subtilis* can balance immune response and tolerance. This study provides dendritic cell (DC)-specific transcriptome data in grass carp by *Bacillus subtilis* stimulation, allowing us to illustrate the molecular mechanism of the DC-mediated immune response triggered by probiotics in grass carp.

## 1. Introduction

Probiotics are a kind of “live microorganism” that have a health benefit on their host [1]. Many studies have demonstrated that they can modulate the host homeostatic immune level and alleviate the inflammatory response caused by specific pathogens [2,3,4]. For instance, *Bifidobacterium longum* has been shown to have a beneficial impact on the immune response by inducing the proliferation of regulatory T cells. It has also been shown to reduce the expression of the pro-inflammatory cytokines tumor necrosis factor-α (TNF-α) and interleukin (IL)-6 in chronic fatigue syndrome (CFS) patients after six-eight weeks of clinical therapy [5,6].

Dendritic cells (DCs) are the most efficient antigen-presenting cells (APCs) and play a central role in regulating the immune response and tolerance [7]. It is widely agreed that DCs can sense and capture antigens from the micro-environment via pattern recognition receptors (PRRs) including Toll-like receptors (TLRs) and C-type lectin receptors (CLRs), which recognize pathogen products called pathogen-associated molecular patterns (PAMPs) [8,9]. Many researchers have reported the effect of probiotics on the immune function of dendritic cells. *Lactobacillus casei* Shirota (LcS) from healthy donors and ulcerative colitis (UC) patients was used to examine the modulation effect on DCs [10,11]. The results suggested that DCs from UC patients have a different expression model in various immune-related molecules when compared with a control group. The production of interleukin (IL)-4 was increased and the expression of interleukin (IL)-22 and interferon (IFN)-γ was down-regulated. However, due to the lack of specific molecule markers of DCs in teleosts, no studies have reported the modulation effects of probiotics on DCs in fish.

Grass carp (*Ctenopharyngodon idella*) is an extensively cultured aquaculture species in China and Southeast Asia. However, the grass carp industry has been threatened by the increasing prevalence of various diseases associated with high mortality [12,13]. The use of chemical drugs and antibiotics in fisheries is facing increasingly stringent restrictions, so the application potential of probiotics in aquaculture is increasing [14,15]. Many studies have demonstrated that probiotics can help the host to combat diseases, accelerate size and weight growth, and, in some cases, act as alternative antimicrobial compounds as well as stimulating the immune response of the host [16,17]. As a commonly used probiotic in aquaculture, *Bacillus subtilis* has demonstrated the ability to metabolize and regulate the intestinal microecology, improve intestinal digestive ability, enhance host immunity, and promote growth [18,19]. However, the regulatory molecule mechanism of *Bacillus subtilis* on the immune system remains unclarified.

Recent studies revealed that probiotic strains differentially modulate the DC expression of cytokines [20,21,22]. *Lactobacillus* can down-regulate IL-12 expression in lipopolysaccharide (LPS)-induced intestinal DC [23], and *Enterococcus faecalis* inhibits the expression of IL-4 and costimulatory molecular CD83, CD86 to reduce the number of mature DCs [24]. In our experiment, we isolated and identified the DCs from white blood cells (WBCs) of grass carp spleen and head kidney, and collected the transcriptome data of dendritic cells following *Bacillus subtilis* stimulation. The key signaling pathways and genes of the immune responses were described and quantitative real-time polymerase chain reaction (qRT-PCR) was used to examine the gene expression changes in RNA sequencing (RNA-seq). These results provide more profound insights into the molecular mechanism by which *Bacillus subtilis* regulate the immune system in grass carp as well as extending the application of probiotics in aquaculture.

## 2. Results

### 2.1. Isolation and Culture of Dendritic-Like Cells in Grass Carp

Due to the lack of tissues such as bone marrow in fish, the mammalian dendritic cell culture protocol was modified [25], and the primary culture was conducted with WBCs from the spleen and head kidney of grass carp. In the early stage of culture, the WBCs were mainly composed of monocytes, macrophages, lymphocytes, and granulocytes (Figure 1a). After 7–10 days of culture, larger, irregular, non-adherent, and dendritic-like cells were present in the medium (Figure 1b). These cells assumed a dendritic morphology suggestive of the DCs of mammals (Figure 1c).

The mammalian dendritic cell enrichment density separation method was used to enrich these cells. The results of the flow forward/side scatter profile analysis indicate that this method can effectively enrich grass carp dendritic cells (gDCs) (Appendix A). The Giemsa stain suggests that the enriched cells were a homogenous population with irregular-shaped nuclei, similar to cells seen in mammalian dendritic cells (mDCs) cultures (Figure 1d). These features are similar to DCs observed in rainbow trout and zebrafish [26,27].

### 2.2. Identification of Biological Function and Expression of Functional Molecular Markers in Dendritic Cells (DCs)

The most characteristic feature of dendritic cells is their remarkable ability to activate naive T cells. Although there is a lack of antibodies to isolate CD4^+^ T cells from grass carp, we can still demonstrate this feature by using the primary mixed lymphocyte reaction (MLR), the gold standard of APC function [26]. Our result suggests that enriched cells mixed with allogeneic responders could indeed stimulate division in a dose-dependent manner (Figure 2). Our data demonstrate that the enriched cells were the functional equivalents of mDCs in T cell activation. Therefore, we decided to further characterize these cells based on other authorized functions of mDCs.

After the antigen had been captured, DCs migrated to the T-lymphocyte enrichment area in vivo. We used the transmembrane migration assay to assess the migration ability of the gDCs. The results indicated that gDCs exposed to lipopolysaccharide (LPS), *A. hydrophila*, and *Bacillus subtilis* have different migration patterns (Figure 3). When exposed to LPS and bacteria in the migration assay, gDCs actively migrated through the pores in the membrane towards LPS and *A. hydrophila* (*p* < 0.05), but not significantly towards *Bacillus subtilis*.

In mammals, the unique activation program of DCs can be triggered by exposure to LPS and other antigens, but whether this mechanism also exists in teleosts is still uncertain. We have successfully detected the expression of CD83, CD80/86, and major histocompatibility complex class II molecules (MHC II) in grass carp in previous studies in our laboratory. These cell-functional molecular markers have been proposed and are widely used to survey the maturation processes of dendritic cells in mammals. We evaluated the expression of these molecular markers in LPS or *Bacillus subtilis* stimulated dendritic cells. Compared to the control group, the expression levels of *CD83* and *CD80/86* in the LPS-stimulated group and *Bacillus subtilis* stimulated group significantly increased, and the expression of *MHC-II* increased in the LPS-stimulated group but not in the *Bacillus subtilis* stimulated group (Figure 4). These data suggest that gDCs and mDCs share similar maturation characteristics [28].

Notably, the results of these experiments successfully demonstrate that the cells isolated from grass carp immune-related tissue have functional homology to mammalian dendritic cells. These results suggest that a systematic analysis is a viable way of clarifying the regulatory molecular mechanism of probiotic *Bacillus subtilis* of dendritic cells in grass carp.

### 2.3. Transcriptome Profiling of DCs in Grass Carp

To ensure the quality of transcriptome information analysis, raw reads were filtered and clean reads were obtained. A total of 72.47 Gb of clean data including 483.14 million read were generated from the three libraries (control group (CG), experiment group 4 h (EG4h), experiment group 18h (EG18h)). Next, we used Hisat 2 (v2.0.1) software to map clean reads to the grass carp genome [29] references and their average mapping coverage percentages in the three libraries were 93.38% (CG), 93.50% (EG4h), and 92.66% (EG18h). The detailed data (Table 1) suggested good coverage of the assembled transcripts by the sequencing read.

In this study, the expected number of fragments per kilobase of transcript sequence per millions base pairs sequenced was set as 1 (FPKM > 1) to define the expressed genes, and the FPKM distribution was visualized as a Violin graph (Appendix A).

### 2.4. Analysis of Differentially Expression Genes (DEGs)

In order to compare gene expression, we normalized expression values using DESeq, and then centered and clustered the values. A |log2(Fold Change)| > 2 and an adjusted *p* value < 0.05 were used as the threshold values to determine differences in gene expression. The three libraries were named CG, EG4h, and EG18h according to their treatments. The centered normalized values were visualized as heat maps and divided into three panels with one for each time point (Figure 5a). The results of the principal component analysis (PCA) showed that the transcript model of CG was different from those of EG4h and EG18h, the latter two transcript models being more similar (Appendix A). The differentially expression genes in each group were visualized as volcano plot maps (Figure 5b). We compared the number of differentially expressed genes among different groups (Figure 5c) to analyze the similarities and differences between different transcriptomes. The comparative transcriptome analysis showed that 1419 transcripts were uniquely different between EG4h and CG, and 559 transcripts were uniquely different between EG18h and EG4h.

### 2.5. Gene Ontology (GO) Analysis and Kyoto Encyclopedia of Genes and Genomes (KEGG) Pathway of DEGs

The gene ontology (GO) classification system was used to determine the possible DEG functions. We compared these three groups and the results showed that the major functional groups had some similarities between the different group (Figure 6a–c). For example, when comparing EG4h and CG, the most enriched groups were “immune response”, “response to stimulus”, “immune system process” and “cytokine receptor binding”; in the EG18h between EG4h group, the major enriched groups were “phospholipid catabolic process”, “immune response”, “response to stimulus”, “cytokine receptor binding”, “G-protein coupled receptor binding”, “chemokine activity”, and “cytokine activity”. These data suggest that *Bacillus subtilis* stimulation mainly affects the immune function of DCs. In addition, cytokines may play an important role in the response of dendritic cells to *Bacillus subtilis* stimulation as the DEGs were continuously enriched in this functional group during stimulation.

The Kyoto Encyclopedia of Genes and Genomes (KEGG) database simulates the functional annotation of the cells or the organism by sequence similarity and genome information [30]. We mapped 2237 DEGs to 148 KEGG pathways in EG4h compared with CG; 981 DEGs to 124 KEGG pathways in EG18h compared with CG; and 1299 DEGs to 141 KEGG pathways in EG18h compared with EG4h. To distinguish the most affected pathways after *Bacillus subtilis* stimulation, KEGG enrichment analysis (corrected *p*-value < 0.05) was performed. We found several pathways that were significantly enriched during the stimulus (Table 2). Notably, the results suggest that the “cytokine–cytokine receptor interaction” pathway was the only pathway that was significantly enriched at both time point (*p* < 0.01), which is consistent with the results of the GO analysis. Therefore, the cytokine-associated genes were verified in the following experiments to clarify their specific roles in response to *Bacillus subtilis* stimulation.

### 2.6. Quantitative Polymerase Chain Reaction (qPCR) Validation and Analysis of Cytokine-Associated Genes

In order to verify the correctness of the RNA-seq data, several genes were selected for RT-PCR analysis. These fold change values of RNA-seq were highly consistent with the qPCR data for all of the selected genes (Figure 7a,b). In addition, to characterize the specific role of the cytokine-cytokine receptor pathway during the *Bacillus subtilis* stimulation, we selected eight cytokine-associated genes (*IL-1β*, *IL-4*, *IL-6*, *IL-8*, *IL-10*, *IL-12*, *TNF-α*, and transforming growth factor (*TGF)-β*) for further analysis. At different time points, the expression of pro-inflammatory factors (IL-1β, IL-6, IL-8, IL-12, and TNF-α), and anti-inflammatory factors (IL-4, IL-10, TGF-β) were both significantly increased (Figure 7c,d). The results showed that *Bacillus subtilis* not only induced dendritic cells to activate the immune system, but also inhibited excessive immunity and maintained the dynamic balance of immunity.

## 3. Discussion

Dendritic cells play a central role in modulating innate immunity and adaptive immunity, especially regarding immune initiation and tolerance [7,31,32]. In our previous study, we established a model of intestinal inflammation in grass carp by pathogenic *A. hydrophila* infection and revealed that *Bacillus subtilis* can effectively restrain inflammation responses [33]. The species was shown to safeguard the integrity of intestinal villi and tight junction structure and restrain *A. hydrophila*-induced down-regulation of tight junction (TJ) proteins zonula occludens-1 (ZO-1). Thus, the transcriptome sequencing of DCs in grass carp will expand the knowledge of *Bacillus subtilis*-host interactions and illuminate the regulatory molecule mechanism of *Bacillus subtilis* to the immune system.

DCs are heterogeneous cells and have a series of subsets according to their phenotype and functions [31]. Due to the lack of specific antibodies, few studies have described the phenotypes of DCs in teleost. Recent studies suggest that DCs in fish and mammalian DCs are highly conserved in phenotypic molecules [34]. Costimulatory molecule CD83 and CD80/86 have been used for the identification of rainbow trout DCs [26,35]. IL-12p40, and MHC-II have also been deemed as biomarkers to evaluate DCs in zebrafish [27]. Our experiments confirmed this conclusion. After being activated through LPS, gDCs showed high expression levels of *CD83* and *CD80/86*, and expression of the major histocompatibility complex class II molecule was also increased. These results suggest that gDCs and other teleost DCs exhibit conserved immunophenotype molecules similar to those of their mammalian counterparts, which may be helpful to realize the evolutionary history of DCs.

In this study, RNA-seq was used to evaluate the differential transcription of the genes of *Bacillus subtilis* stimulated gDCs at different time points. To facilitate the global analysis of DEGs, we used GO analysis [36] to determine the major functional categories that the DEGs were involved in. In this study, the significantly enriched categories of DEGs differed among time points. At 4 h, DEGs involved in the biological process were dominant and those involved in molecular function dominated at 18 h. These results suggest that the effect of *Bacillus subtilis* stimulation on gDC is associated with a complex biological process. In *Bacillus subtilis*-stimulated gDCs, several up-regulated genes were significantly enriched in the GO terms of cytokine activity, cytokine receptor binding, and chemokine receptor binding, which are related to immune responses. It is widely agreed that cytokines and chemokines play important roles in immunoregulation; and induce the differentiation of immune cells [37,38,39]. For instance, IL-10 and IL-27 can activate and induce the generation of tolerogenic DCs via the TLR2/MyD88-dependent pathway [40,41,42]. Our results may help to further understand the change of immune responses in teleosts over time and the specific molecular mechanisms by which probiotics regulate DCs.

In order to identify the biological pathways that were active in the stimulation, a KEGG database was used to map all of the DEGs. Most enriched KEGG pathways were related to metabolism and immune responses, suggesting that *Bacillus subtilis* stimulus could enhance the metabolism and regulate the immune functions of gDCs, which is similar to the effect of probiotics in mammals [43,44]. It is worth noting that the cytokine–cytokine receptor pathway was significantly enriched during the stimulus. As is widely known, cytokines play an important role in immune regulation [45,46]; and can be divided into pro-inflammatory cytokines and anti-inflammatory cytokines which balance the immune response [47,48,49]. In our study, eight cytokines (five pro-inflammatory cytokines (IL-1β, IL-6, IL-8, IL-12, and TNF-α) and three anti-inflammatory cytokines (IL-4, IL-10, and TGF-β)) were chosen to evaluate the effects on cytokine signaling pathways during stimulation. The RT-PCR results suggest that they were all highly expressed at both time points, especially IL-10 and IL-12. Some studies have described that different strains of probiotics can modulate IL-10 and IL-12 secretion differently. *Lactobacilli* can promote the production of IL-12 effectively and *Bifidobacteria* induce an effect on IL-10 secretion [50,51]. Researchers have investigated the mechanisms of different strains that differentially induce IL-12 and IL-10 production. The results suggested that bacterial teichoic acids act as the key factor for inducing IL-10 production, which is mediated by TLR2-dependent ERK activation [52,53]. The rigid cell wall, which is resistant to intracellular digestion, is also thought to cause the difference between this difference in cytokine responses. *L. casei* can also be transformed from the IL-12-inducing strain to the IL-10-inducing strain in the presence of teichoic acids [1,52]. Therefore, the high expression levels of IL-10 and IL-12 can be explained by the thick cell wall and the presence of bacterial teichoic acids in *Bacillus subtilis*. In addition, the expression of the anti-inflammatory cytokine TGF-β; was also up-regulated and stably maintained. In a previous study, TGF-β demonstrated the ability to inhibit the proliferation of IL-2-induced T cells [54]. As an important negative regulatory molecule in the TLR4 signaling pathway, TGF-β induced apoptosis of various immune cells including B cells and eosinophils [55,56]. However, in our transcriptome data, TLR-4 related genes were not detected. Although the potential reasons for this are not fully clear, possible explanations might include, but are not limited to (1) there being signaling pathways in grass carp other rather than TLR4 signaling pathway; and (2) no transcription of these specific genes was detected due to temporal expression patterns.

As the RNA-seq and qPCR results suggest, the mRNA expression levels of both pro-inflammatory and anti-inflammatory cytokines significantly increased during *Bacillus subtilis* stimulation. Whether this change in transcription level is consistent with the translation level requires further experiments for confirmation. Moreover, based on the complexity of the host internal environment, the response of dendritic cells to *Bacillus subtilis* may be different when compared with the response in vitro [57]. Future studies should focus on the use of specific molecular markers to analyze the immunomodulatory effect of *Bacillus subtilis* on dendritic cells in the specific tissues of grass carp.

In conclusion, dendritic cells in grass carp were isolated and identified for the first time, and we generated the transcriptomes of dendritic cells in grass carp stimulated by *Bacillus subtilis*. GO analysis and KEGG pathway analyses suggested that *Bacillus subtilis* regulates the immune functions of DCs by affecting the expression of cytokines. The RT-PCR results further confirmed that *Bacillus subtilis* could up-regulate the expression of both pro-inflammatory and anti-inflammatory cytokines. Our results provide a deep insight into the immune responses of dendritic cells to *Bacillus subtilis* stimulation and will be useful in refining our knowledge of the regulatory molecular mechanism of probiotic to the immune responses in fish.

## 4. Materials and Methods

### 4.1. Experimental Animals

Grass carps were acquired from Hubei Bairong Improved Aquatic Seed Co. Ltd. (Huanggang, China). The fish were maintained in flow-through tanks at 25–28 °C. Fish were fed commercial grass carp chow (Haid Group, China) at a quantity of 1.5–2% of their body weight twice per day and were acclimated for two weeks before the experiment. The fish were euthanized in water with 3-aminobenzoic acid ethyl ester methanesulfonate (MS-222) before the experiment. All experiments were approved by the Animal Ethics Committee of Huazhong Agricultural University on 15 March 2017. The ARRIVE Guidelines for animal research were followed in this experiment (Additional file 1: ARRIVE checklist).

### 4.2. Bacterial Strains

As described above [33], *B. subtilis* Ch9 and *A. hydrophila* strains were acquired from the Laboratory of Aquatic Animal Medicine, College of Fisheries, Huazhong Agricultural University. *B. subtilis* was incubated on a Luria–Bertani (LB) plate at 37 °C for 24 h, and then we picked single colonies and inoculated them into liquid LB medium at 37 °C for 14 h. *A. hydrophila* was incubated at 28 °C for 14 h, and underwent centrifugation at 5000 rpm for 10 min and was then washed twice in phosphate-buffered saline (PBS). Bacteria cells were adjusted to the appropriate concentration in L-15 medium before the experiment. *B. subtilis* Ch9 suspension was inactivated by ultraviolet irradiation for 30 min. Ultraviolet light (UV)-killed bacteria were inoculated on LB plates to examine the activity.

### 4.3. Isolation and Enrichment of Dendritic-Like Cell

The 1000–2000g grass carp were euthanized with overdosed MS-222 (Sigma-Aldrich, MO, USA). Spleens and head kidneys were collected under sterile conditions. Excised tissue was immersed in supplemented media containing L-15 media (Hyclone, Logan, USA) with 10% fetal bovine serum (Sciencell, CA, USA), 1% penicillin-streptomycin (PS; 10,000 U/mL) and fungizone (0.25 mg/mL). Tissues were cut into small pieces and forced through a 70 mm cell strainer (BD, NJ, USA) gently, and a single cell suspension (lymphocyte) was gained. The cell suspension was layered over 1.077 g/mL Percoll at a 1:1 ratio, and then spun at 800× *g* for 20 min and washed twice in PBS. Cells were counted and adjusted to 5 × 10^6^ cells/mL in supplemented media and inoculated into 6-well plates (Corning, NY, USA). After 7–10 days of culture at 28 °C, non-adherent cells were harvested for further experiments.

To enrich the dendritic cells, the non-adherent cells in the media were collected. The cell suspension was layered over nycoprep/one-step monocytes (Sigma-Aldrich, MO, USA) and underwent centrifugation at 2000 rpm for 30 min at room temperature. Enriched cells were removed and washed twice in PBS before further use.

### 4.4. Mixed Leukocyte Reaction and Migration Assay

Spleens were taken from the euthanized grass carp aseptically and forced through a 70 mm cell strainer gently to gain a single cell suspension. The cell suspension was layered over 1.077 g/mL Percoll at a 1:1 ratio, and then spun at 800× *g* for 20 min at room temperature. After centrifugation, the buffy coat containing the lymphocyte fraction was removed and washed twice in PBS. Stimulator cells (gDCs) were resuspended and adjusted to 1 ×10^6^ cells/mL in L-15. Responder cells were labeled with 10 mM CFSE (Invitrogen, CA, USA) for 10 min. In order to stop labeling, we added five volumes of cold L-15 media into the responder cells, and then washed them twice in PBS. Cells were resuspended to 1 × 10^6^ cells/mL in MLR medium. Stimulators were then planted in 100 μL MLR medium at 1 × 10^5^ cells per well in 12-well plates (Corning, NY, USA). A total of 1 × 10^5^ responder cells were then added to all wells, resulting in a stimulator to responder ratio of 1:1 (ratios of 1:2, 1:4, and 1:8 were initially used). Control group wells contained 2 × 10^5^ responder cells alone. After three days (determined to be an optimum incubation time experiment carried out to day 5) of incubation, cells were resuspended in PBS for flow cytometry analysis (Beckman Coulter, CA, USA). Control groups (responder cells alone) were used to set the gate for analysis as the background (gate set at approximately 1.8% dividing cells).

The migratory ability of gDCs was evaluated by using a 12-well Transwell culture plate (Corning, NY, USA). In this study, membranes with 3 μm pores were chosen to allow active migration as well as to decrease the passive transfer to a minimum range. For each treatment group, 1.5 mL of 0.5% fetal bovine serum (FBS)/L-15 media was placed in the lower compartment and lipopolysaccharide (LPS; 5 μg/mL) and live *A. hydrophila* and *Bacillus subtilis* ch9 (strains at a multiplicity of infection (MOI) of 0.1 and 1) were used as the treatment groups. The control group was composed of 0.5% FBS/L-15 media alone. gDCs (500 μL of 1 × 10^6^ cells/mL) were resuspended in 0.5% FBS/L-15 media and added to the upper compartment. After that, the plate was incubated at 28°C for migration. After 4 h of incubation, by gently swabbing a cotton bud against the membrane, the cells remaining in the upper compartment were removed. The cells attached to the lower compartment of the membrane were detached with trypsin (10 μg/mL for 2 min). Cells were then stained with 4’,6-diamidino-2-phenylindole (DAPI) (2.5 μg/mL for 15 min) and counted on a hemocytometer using fluorescent microscopy.

### 4.5. Functional Molecular Markers Expression Analysis

LPS (5 μg/mL) and UV-killed *B. subtilis* Ch9 (MOI = 100) were co-incubated with dendritic cells in 12-well plates. After 12 h of stimulation, cells were collected to extract RNA for the expression of functional molecular markers analysis.

### 4.6. Preparation of Cell Samples Stimulated by Bacillus subtilis

UV-killed *B. subtilis* Ch9 was added to the dendritic cell medium in 12-well plates at a MOI of 100. We collected cells at 0 (as the control group), 4, and 18 h, and extracted total RNA. Three technical replicates (individual wells) were pooled into one biological replicate. Total RNA was extracted from the samples with Trizol (Invitrogen, CA, USA) in accordance with the manufacturer’s protocols.

### 4.7. cDNA Library Preparation and Illumina Sequencing

After RNA extraction, poly(A)-containing mRNAs were purified using oligo(dT)-attached magnetic beads. Equal amounts of RNA isolated from different samples were mixed. Using the fragmented RNA as the template, first-strand cDNA was synthesized using random primers. Second-strand cDNA was synthesized using a buffer containing DNA polymerase I, RNase H, and dNTPs. cDNA fragments were ligated to adapters after the end-repair process. These products were developed into a cDNA library after purification and enrichment. Finally, the complete library was sequenced by Novogene Bioinformatics (Tianjin, China).

### 4.8. Analysis of DEGs

Differentially expressed genes were analyzed using the DEGseq package in R. Stringent criteria of a *p* adjusted value < 0.05 and |log2(Fold Change) | > 2 were used to define differentially expressed genes. Biological functions and pathways involving these DEGs were determined using the GO annotation system and KEGG database, respectively. GO terms with a corrected *p*-value ≤ 0.05 were considered as significantly enriched for DEGs. Pathways with a corrected *p*-value ≤ 0.05 were considered as significantly enriched pathways for DEGs

### 4.9. Real-Time Quantitative Polymerase Chain Reaction (qRT-PCR) Analysis

To examine the expression of cytokine-associated genes induced by *Bacillus subtilis* and to verify the transcriptome data, we used qRT-PCR analysis. RNA extraction and cDNA synthesis were conducted as described above [33]. The primers used for qRT-PCR were designed based on the assembled gene sequences using Premier 5 software (Premier Biosoft International, CA, USA) (Table 3). The assay was analyzed in three independent replicates.

### 4.10. Statistical Analysis

All statistical analyses were performed using SPSS 18.0 software (SPSS. Inc., IL, USA). Statistical differences were evaluated using the Student’s *t*-test for unpaired samples. Analysis of variance (ANOVA) with Tukey post-tests were used to compare the differences between multiple groups. Differences between experimental groups were considered significant at a probability of error of *p* < 0.05. Data are presented as the mean ± standard deviation (SD).

### 4.11. Data Availability

The datasets generated and/or analysed during the current study are publicly available at NCBI–SRA (www.ncbi.nlm.nih.gov/sra) accession: SRR8402455, SRR8402456, SRR8402457, SRR8402458 SRR8402459 SRR8402460, SRR8402461, SRR8402462 and SRR8402463.

## 5. Conclusions

In this study, we isolated and identified dendritic cells in grass carp. We demonstrated that these cells are highly conserved as compared with their counterparts in mammals; and share conserved immunophenotype molecules similar to those of their mammalian counterparts during LPS stimulation. We combined the transcriptome and qRT-PCR methods to analyze the effect of *B. subtilis* on DCs. Our results suggest that the immune function of DCs is mainly affected by the regulation of cytokine-related pathways during stimulation. In addition, it significantly induces the mRNA expression levels of both pro-inflammatory (IL-1β, IL-6, IL-8, IL-12, and TNF-α) and anti-inflammatory cytokines (IL-4, IL-10, and TGF-β). Our study provides a huge volume of data that can be used to explore the molecular mechanism of grass carp DCs’ response to *Bacillus subtilis* stimulation and may be helpful in understanding the regulation of host immunity by probiotics in aquaculture.

## Figures and Tables

**Figure 1 ijms-20-00389-f001:**
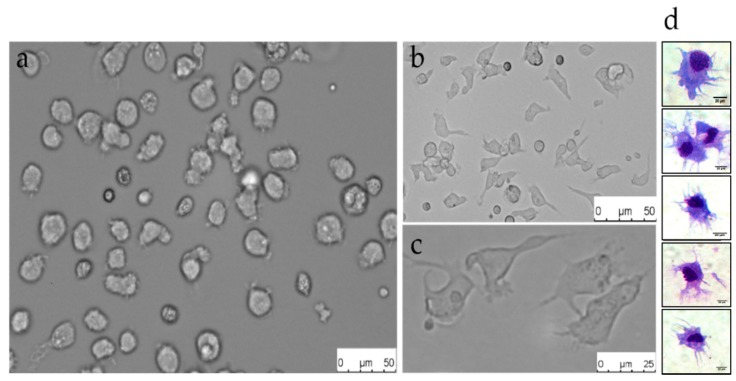
Culture and staining observation of grass carp dendritic cells (gDCs). (**a**) Isolated mononuclear cells at day 1 (scale bar = 50 μm). (**b**) Cultures showing non-adherent cells with dendritic morphology (scale bar = 50 μm). (**c**) Enriched cells with typical dendritic morphology (scale bar = 25 μm). (**d**) Dendritic cells were stained using Giemsa showing typical dendritic morphology on these cells (scale bar = 20 μm). Data are representative of at least three independent experiments (*n* > 3).

**Figure 2 ijms-20-00389-f002:**
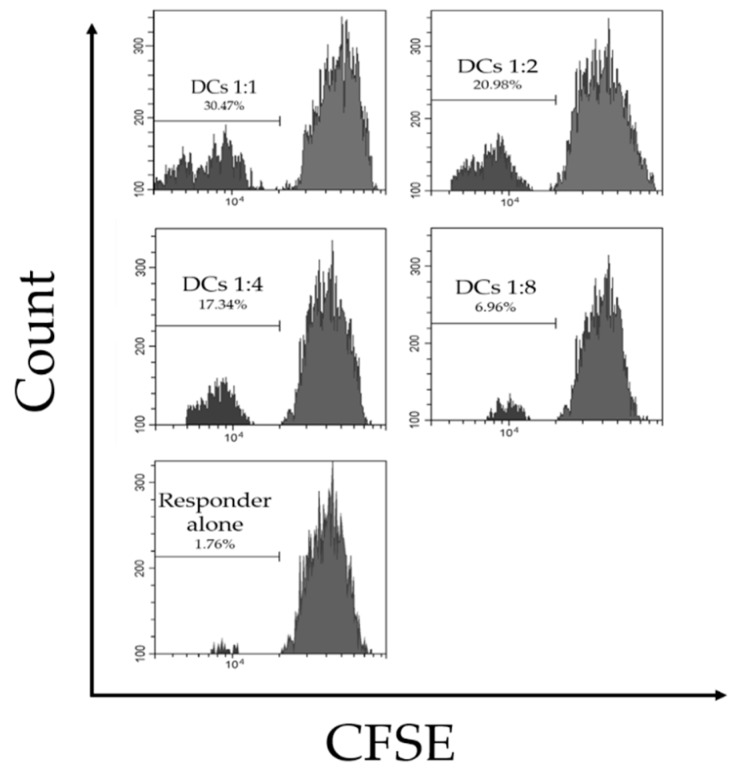
Mixed lymphocyte reaction (MLR) experiments of dendritic cells in grass carp. Grass carp dendritic cells were cultured with 5-(and-6)-carboxyfluorescein diacetate, succinimidyl ester (CFSE) stained spleen responders in a primary allogeneic MLR. The ratios of gDCs to responders were 1:1, 1:2, 1:4, and 1:8. The numbers represent the percentages of dividing responders. Data are representative of four independent experiments, each using stimulators and responders from multiple individual fish.

**Figure 3 ijms-20-00389-f003:**
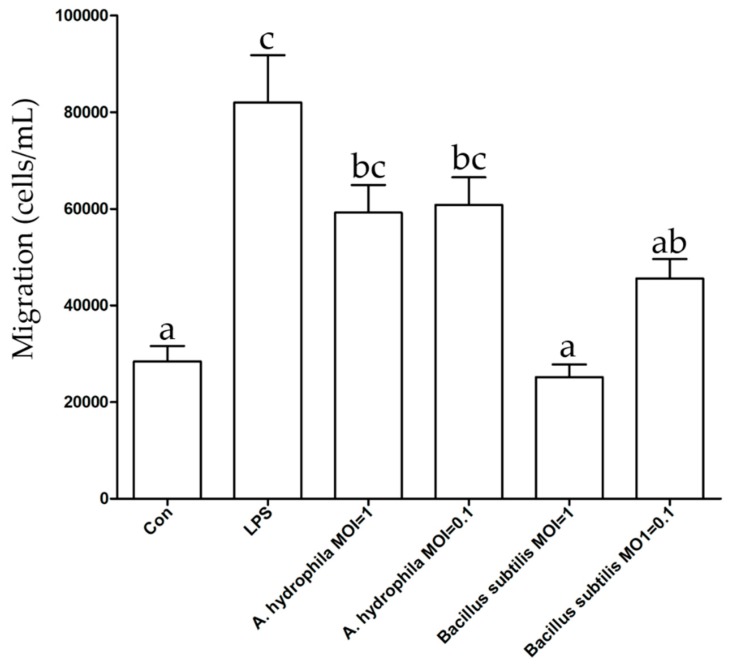
Migration analysis of gDCs towards different treatments. Pairwise group comparisons were conducted using analysis of variance (ANOVA) with the Tukey post-test. *p* < 0.05 indicated statistically significant differences between groups. Data are expressed as mean ± standard deviation (SD) (*n* = 3). Shared letters above bars indicate the groups were not statistically different from one another (*p* > 0.05), while differing letters indicate significantly different mean values (*p* < 0.05).

**Figure 4 ijms-20-00389-f004:**
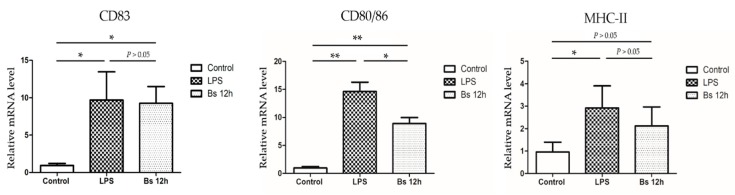
Changes in the expression of functional molecular markers (*CD83*, *CD80/86*, and *MHC II*) in dendritic cells under lipopolysaccharide (LPS) or *Bacillus subtilis* stimulation. The mRNA levels measured by quantitative polymerase chain reaction (qPCR) are presented as the relative fold change normalized against the housekeeping β-actin gene. All the three independent experiments had the same trend on functional molecules by quantitative real-time PCR (qRT-PCR). Data are expressed as mean ± SD (*n* = 3). * *p* < 0.05 and ** *p* < 0.01 indicate statistically significant differences as compared with the control group.

**Figure 5 ijms-20-00389-f005:**
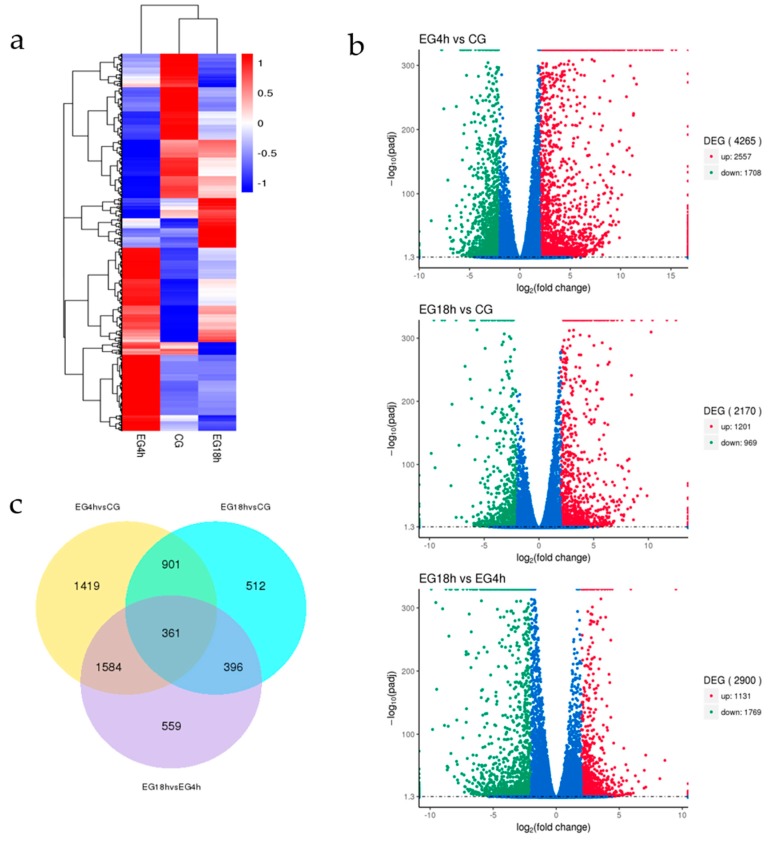
Differentially expressed gene analysis. (**a**) Differentially expressed genes (DEGs) on the clusters were performed by two-dimensional hierarchical clustering. (**b**) Differentially expressed genes from three groups were visualized as Volcano plot maps. Up-regulated and down-regulated transcripts in EG4h compared to control group (CG), EG18h compared to CG and EG18h compared to EG4h were determined, respectively. (**c**) The Venn diagram suggests overlaps between the different expression transcripts within each compared group.

**Figure 6 ijms-20-00389-f006:**
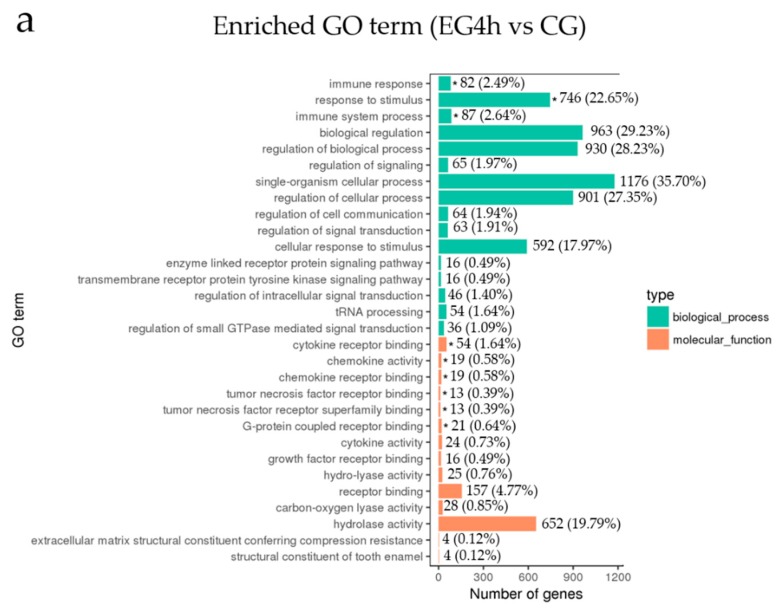
Gene ontology (GO) classification of the DEGs. Differentially expressed gene annotation hits were from the GO databases. We listed the most 30 enriched GO terms obtained between EG4h vs CG (**a**), EG18h vs CG (**b**), EG18h vs EG4h (**c**). The number of DEGs and their percentage in each GO terms were shown in the graph. * means significantly enriched (*p* < 0.05).

**Figure 7 ijms-20-00389-f007:**
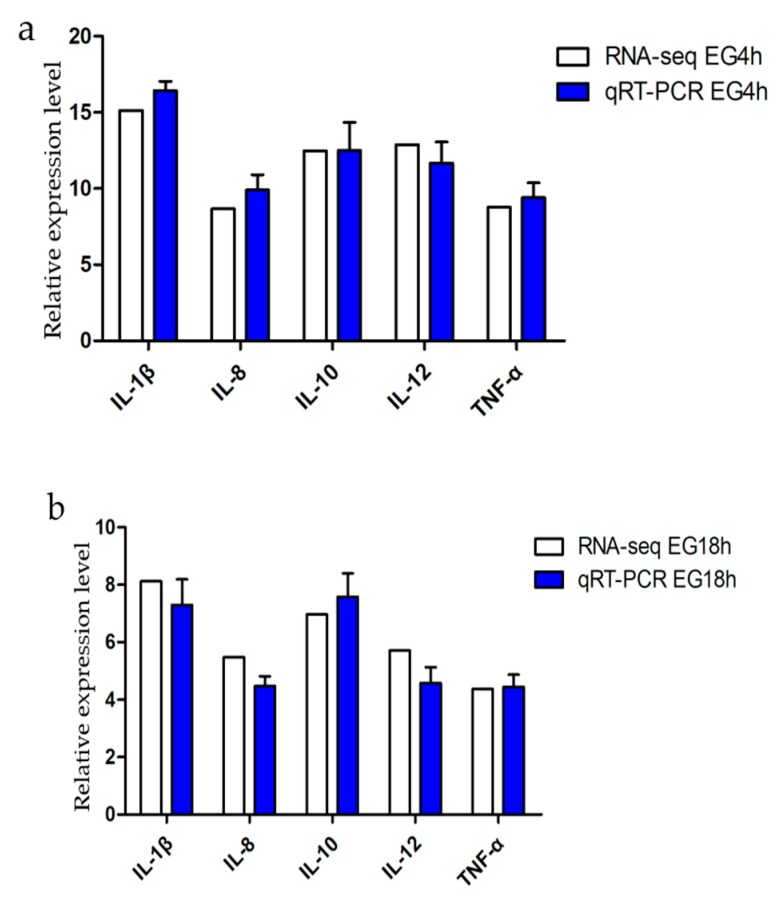
Validation of RNA-seq data by qPCR and the mRNA expression level of immune-related cytokines after *Bacillus subtilis* stimulation. (**a**) The results of qPCR validation of EG4h, compared with the RNA-seq data of EG4h. The expression levels of five genes were detected by qRT-PCR (blue column) and compared with the results of RNA-seq (white column). (**b**) The results of qPCR validation of EG18h, compared with the RNA-seq data of EG18h. The expression levels of five genes were detected by qRT-PCR (blue column) and compared with the results of RNA-seq (white column). (**c**) The expression of five pro-inflammatory cytokine genes (*IL-1β*, *IL-6*, *IL-8*, *IL-12*, and *TNF-α)* increased during the stimulation. (**d**) Three anti-inflammatory cytokine genes (*IL-4*, *IL-10*, and *transforming growth factor* (*TGF)-β*) increased at both time point. All the three independent experiments had the same trend on cytokines by qRT-PCR. Data are expressed as mean ± SD (*n* = 3). * *p* < 0.05, ** *p* < 0.01 compared with the control group.

**Table 1 ijms-20-00389-t001:** Mapping analysis of clean data with reference genomes of grass carp (three technical replicates (individual wells) were pooled into one biological replicate, each sample represent one biological replicate).

Samples Name	CG-1	CG-2	CG-3	EG4h-1	EG4h-2	EG4h-3	EG18h-1	EG18h-2	EG18h-3
Total reads	61,979,920	52,440,886	54,798,236	56,721,626	58,936,752	49,222,662	47,365,976	53,178,034	48,492,348
Total mapped	57,889,313 (93.4%)	48,940,763 (93.33%)	51,182,959 (93.4%)	53,222,399 (93.83%)	55,091,143 (93.48%)	45,866,160 (93.18%)	43,997,429 (92.89%)	48,960,669 (92.07%)	45,107,032 (93.02%)
Multiple mapped	2,198,393 (3.55%)	1,855,491 (3.54%)	1,994,290 (3.64%)	2,004,745 (3.53%)	2,081,938 (3.53%)	1,756,479 (3.57%)	1,805,018 (3.81%)	1,955,890 (3.68%)	1,879,391 (3.88%)
Uniquely mapped	55,690,920 (89.85%)	47,085,272 (89.79%)	49,188,669 (89.76%)	51,217,654 (90.3%)	53,009,205 (89.94%)	44,109,681 (89.61%)	42,192,411 (89.08%)	47,004,779 (88.39%)	43,227,641 (89.14%)
Reads map to ‘+’	27,761,963 (44.79%)	23,472,551 (44.76%)	24,515,223 (44.74%)	25,531,775 (45.01%)	26,423,909 (44.83%)	21,987,005 (44.67%)	21,040,234 (44.42%)	23,426,471 (44.05%)	21,557,729 (44.46%)
Reads map to ‘−’	27,928,957 (45.06%)	23,612,721 (45.03%)	24,673,446 (45.03%)	25,685,879 (45.28%)	26,585,296 (45.11%)	22,122,676 (44.94%)	21,152,177 (44.66%)	23,578,308 (44.34%)	21,669,912 (44.69%)

**Table 2 ijms-20-00389-t002:** The significant enrichment pathways at different time points. The pathway ID was obtained from the Kyoto Encyclopedia of Genes and Genomes (KEGG) database. A corrected *p*-value < 0.05 were used to identify significantly enriched pathways.

Pathways	Pathways ID	Input Numbers at 4 h	Corrected *p*-Value at 4 h	Input Numbers at 18 h	Corrected *p*-Value at 18 h
DNA replication	dre03030	23	0.004062352	7	0.374206007
Cytokine-cytokine receptor interaction	dre04060	56	0.004163873	45	4.54045 × 10^−9^
Extracellular matrix (ECM)-receptor interaction	dre04512	26	0.1689689	19	0.006915202

**Table 3 ijms-20-00389-t003:** Primers used in the experiment.

Gene	Accession No.	Annealing Temperature (°C)	Primer
*IL-1β*	JQ692172.1	57.1	Fwd:5′-AGAGTTTGGTGAAGAAGAGG-3′ REV:5′-TTATTGTGGTTACGCTGGA-3′
*IL-4*	KT445871	55.9	Fwd:5′-CTACTGCTCGCTTTCGCTGT-3′ REV:5′-CCCAGTTTTCAGTTCTCTCAGG-3′
*IL-6*	KC535507.1	62.3	Fwd:5′-CAGCAGAATGGGGGAGTTATC-3′ REV:5′-CTCGCAGAGTCTTGACATCCTT-3′
*IL-8*	JN663841.1	60.3	Fwd:5′-ATGAGTCTTAGAGGTCTGGGT-3′ REV:5′-ACAGTGAGGGCTAGGAGGG-3′
*IL-10*	HQ388294.1	61.4	Fwd:5′-AATCCCTTTGATTTTGCC-3′ REV:5′-GTGCCTTATCCTACAGTATGTG-3′
*IL-12*	KF944668.1	59.0	Fwd:5′-ACAAAGATGAAAAACTGGAGGC-3′ REV:5′-GTGTGTGGTTTAGGTAGGAGCC-3′
*TGF-β*	EU099588.1	55.9	Fwd:5′-TTGGGACTTGTGCTCTAT-3′ REV:5′-AGTTCTGCTGGGATGTTT-3′
*TNF-α*	HQ696609	56.0	Fwd:5′-CTTCACGCTCAACAAGTCTCAG-3′ REV:5′-AAGCCTGGTCCTGGTTCACTC-3′
*β-actin*	DQ211096	61.4	Fwd:5′-CCTTCTTGGGTATGGAGTCTTG-3′ REV:5′-AGAGTATTTACGCTCAGGTGGG-3′

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
