# Peer review of "Regulatory Effect of Bacillus subtilis on Cytokines of Dendritic Cells in Grass Carp (Ctenopharyngodon Idella)"

_ijms, 2019, doi:10.3390/ijms20020389_

Reviewer 1 Report

In this manuscript, Chen and co-authors established a method to isolate dendritic cells from grass carp and investigated the transcriptional profile of DCs in response to Bacillus subtilis treatment. They found that Bacillus subtilis can affect the expression of several cytokine related pathways in DCs which may affect DC function. This study not only provides important source data for grass carp DC research but also may be helpful to understand the regulation of host immunity by probiotics in aquaculture. I have the concerns below which should be addressed before publication.

1. Line 25, DEGs should be defined and annotated.

2. Scale bar should be provided for Fig 1d. In addition, the author claimed that the isolated cells are DCs. But this is mainly based on the morphology, any other evidence such as surface markers can be provided to further support this conclusion?

3. The label of X-axis and Y-axis should be provided for the flow data in Fig 2a, the label of Y-axis should be provided for Fig 2b. Moreover, how many times are the experiments repeated for?

The figure legend for Fig 2b should be revised. What does the different letter mean, significant or not?  

4. Line 125, the name for C. Idella should be consistent in the whole text, either C. Idella or grass carp is fine.

5. What is the difference between Fig 6a and b? The figure legend should be revised.

6. A MLR experiment should be performed to directly examine the effect of Bacillus subtilis on DC function.

Author Response

Thank you for your nice comments on our article. Based on your comments, we have made extensive revisions to our previous draft. We submit the revised manuscript as well as a list of changes in this paper. If you have any question about this paper, please don’t hesitate to let me know.

Reviewer 2 Report

General comments that must be addressed:

1. To support the open science movement, and substantiate your final concluding sentence that states ‘our study provides huge volumes of data’, the entire illumina dataset should be uploaded somewhere with open access, and the link to this dataset should be provided within the publication. Please include the raw and cleaned data, including adequate annotation for other researchers to easily understand and use the results.

2. Please correct the grammar and improve the general phrasing throughout the entire manuscript with the help of a native English speaker or professional editor.

3. Please ensure you have defined each acronym the first time it is used in the manuscript (e.g. MLR, mDCs à is this mouse or monocyte derived DCs, and from which species / tissue).

4. If flow cytometry antibodies that can detect surface MHC-II and CD86/83/80 on grass carp cells are available, please perform a staining to demonstrate that the surface expression of these markers on enriched gDCs is modulated at the protein level in a similar way to the transcriptomic changes you detected. Include a probiotics group to compare to LPS stimulation also.

5. Every figure legend and table explanation must include the number of experimental repeats (and number of technical replicates if these were performed)

6. If you have Illumina data from other cell subsets (non DCs, or gDCs at different times during enrichment) that was generated during the same experiments as those in the publication, please include that as supplementary data as a control or reference point to put the gDC expression levels into perspective

Other suggestions

Introduction:

Mention which species you obtained white blood cells from

Results

Figure 2: in part A, label the axes. In part B, please clarify the statistical comparisons being made because it is unclear what the different letters a,b,c refer to (also in general it is advisable to not use alphabet letters within a figure because letters are already being used to label the different figures).

Section 2.2. Because you have only detected transcriptomic molecules from lysed cells you cannot know whether these molecules are expressed at the surface or intracellularly. Therefore remove the word surface from the subheading. You could replace ‘surface’ with ‘functional’ or ‘co-stimulatory’, which would be more informative. This applies to other sections of the manuscript as well, e.g. methods 4.5.

Figure 3: please show the effects of probiotics on CD83/86/80 and MHC-II as a direct head- to-head comparison with LPS stimulation, and list the fold change difference compared to the control in a supplementary table or within the results text.

Figure 4: many labels need to be much larger in the figure so that they become legible.

Line 107, incorrect because A. hydrophilia response is the same for 1 and 0.1 MOI.

Line 178, major typographical error (duplication of phrase)

Figure 5. use a,b, and c to designate the 3 graphs instead of a1, a2, a3. Improve overall presentation of figure and include additional data such as how many genes in the entire go term, any statistical values, or the percentage of DEGs that were represented in each go term. Delete the heading from each graph (The most enriched GO Terms) and move it somewher into the figure legend. Ensure the full GO Term title is visible.

Discussion

Explain in more detail how the significant GO Terms and Pathways could be used to make new therapies or approaches to improve fish health, or guide new research, e.g. propose additional new hypotheses and how to test these hypotheses, or predict in which scenarios the different probiotics would be most and least useful in fisheries.

Add more references for your statements about techoic acid

Note, the cells could have already partially matured during the enrichment process, and this may have resulted in reduced TLR4. However your suggestion that the gDCs use other TLRs to recognize the probiotics is highly plausible.

Supplementary Figures

Figure S1. Please add values to the x and y axis of the Flow Cytometry plots and state what kind of flow cytometer was used somewhere in the manuscript. It would also be informative to plot a gate on the position of the cells you define as DCs, and state the percentage of this population in the results text for both pre and post-enriched samples. In this way, the reader can also have a rough estimate of whether few or many very large contaminating cells (possibly non-adherent monocytes/macrophages or activated DCs) are present in the enriched culture. Also please explain how you determined purity was more than 80%, as stated in he figure legend.

Methods

4.2 please include a detailed description of the components within your L-15 medium

4.3. Briefly define or describe the ‘head kidney’ in more detail because this is not a commonly used organ. Is it lymphoid-like? What is its function etc?

Also include a few more details about the nycoprep centrifugation step (speed, time, temperature) and which cell layer was removed and how, so that the method can be accurately reproduced by other scientists without having to find the manufacturer’s instructions and guess how you handled the cells at this critical step

4.10. Statistics in major immunology journals do not often use a Duncan test to compare multiple groups. Please justify your choice of this test, or preferably use a more accepted test such as Mann Whitney, or ANOVA with Tukey or Bonferoni post-tests, where appropriate, to demonstrate that your main findings are statistically significant with alternative tests. If needed, the online guide to GraphPad Prism software provides clear instructions on how to choose appropriate statistical tests for different data set.

Optional experiments that would improve the quality of the publication

It would be very informative to know the expression level of CD11c/CD141/CD11b/CD103/CD4/CD8/CD64 if cytometry Abs are available/cross-reactive with carp, on adherent cells and non-adherent cells during different time points of the culture to determine whether the enrichment process activates DCs or enriches certain DC subsets - or just to know what kinds of DCs a carp has! This is important because you may not need to wait until dendritic morphology is seen during the enrichment process to have functional and ready to use cells. By providing your full Illumina dataset as an open resource, researchers may be able to determine whether the gDCs are homologous to cDC1/cDC2 subsets and identify which pathogen or danger receptors are expressed, or other drug targets, and thus help to develop new immune-based medicines for carp.

Author Response

Thank you for your nice comments on our article. Based on your comments, we have made extensive revisions to our previous draft. We submit the revised manuscript as well as a list of changes in this paper. If you have any question about this paper, please don’t hesitate to let me know.

Round  2

Reviewer 1 Report

All my concerns were addressed properly. I support the publication of this manuscript.